# Steam Gasification of Torrefied/Carbonized Wheat Straw for H_2_-Enriched Syngas Production and Tar Reduction

**DOI:** 10.3390/ijerph191710475

**Published:** 2022-08-23

**Authors:** Kejie Wang, Ge Kong, Guanyu Zhang, Xin Zhang, Lujia Han, Xuesong Zhang

**Affiliations:** Engineering Laboratory for AgroBiomass Recycling & Valorizing, College of Engineering, China Agricultural University, Beijing 100083, China

**Keywords:** H_2_-riched syngas, tar elimination, steam gasification, torrefaction, carbonization, wheat straw

## Abstract

Torrefaction/carbonization integrated with steam gasification of agricultural biomass for gas production and tar reduction was not investigated. The aim of this study was to evaluate the influence of the torrefaction/carbonization severity on H_2_-enriched syngas production and tar reduction during steam gasification of wheat straw (WS). The torrefaction/carbonization experiments were initially performed at 220–500 °C to examine the effect of pretreated temperature on the fuel properties of torrefied/carbonized WS. Then, the gasification temperature (700–900 °C) was optimized at 900 °C in terms of gas formation behaviors. Afterward, steam gasification of raw and torrefied/carbonized WS feedstocks was conducted. WS carbonized at 500 °C (WS-500) possessed the highest H_2_ concentration (54.21 vol%) and syngas purity (85.59%), while the maximum H_2_/CO molar ratio (1.83), high carbon conversion efficiency (90.33 C%) and cold gas efficiency (109.24%) were observed for WS torrefied at 280 °C. Notably, the cumulative gas yield, H_2_ yield, and syngas yield respectively reached 102.68 mmol/g, 55.66 mmol/g, and 87.89 mmol/g from steam gasification of WS-500. In addition, the carbonized WS feedstocks, especially WS-500, revealed a lower tar content. Simply put, integrating torrefaction/carbonization with steam gasification provided a novel and effective route to manufacture H_2_-enriched syngas with extremely low tar content from agricultural biomass.

## 1. Introduction

With the growing concerns related to petroleum resource usage, developing renewable energy can essentially solve these threats posed by the energy crisis and climate change [1,2]. Biomass has emerged as a promising candidate instead of petroleum resources for the production of heat, electricity, syngas, liquid fuels, and chemicals [3,4,5,6]. Thermochemical technologies such as pyrolysis and gasification are effective and promising platforms for turning biomass into biochemicals and biofuels [7]. 

Gasification is a variable option to manufacture syngas from biomass for downstream utilization in combustion, Fisher–Tropsch synthesis, syngas fermentation, and so forth [4,8,9,10,11]. Biomass gasification mainly contains several stages: (1) pyrolysis of biomass into volatiles and biochar, (2) intermediate volatiles reacting with gasifying agents (air, oxygen, steam, or their mixtures) through secondary reactions (combustion and reforming), and (3) char reacting with oxygen, steam, and CO_2_ [3,12]. Thus, finding an efficient and cost-effective way to produce high-quality syngas is of great importance for biomass valorization [4]. 

The unfavored characteristics of biomass feedstocks, e.g., high oxygen and moisture contents, low energy, and bulk densities, hydrophilic and tenacious fibrous natures, and heterogeneous compositions [13,14,15,16], result in huge challenges to biomass logistics and downstream thermochemical processes [16,17,18,19]. What is worse, the rigid fibrous structure can make raw biomass difficult to ground into small particles and ensure a complete gasification performance [18]. It is also discerned that the high volatile content and low energy density of biomass can cause large amounts of tar to form during biomass gasification [16]. The undesirable tar formation usually gives rise to plugging, fouling, pipeline corrosion, and catalyst deactivation [20,21,22]. Tar abatement is regarded as one of the primary challenges facing the industrialization of biomass gasification [3]. It is well known that post-gasification treatment often suffers from the high energy penalty and capital costs [3]. Comparably, pretreatment integrated with in situ catalytic cracking is viewed as the ideal way to handle this challenge [3]. 

The common pretreatment approaches are torrefaction, carbonization, hydrothermal pretreatment, steam explosion, etc. [3,23,24,25]. Torrefaction is a low-temperature thermal treatment process that is commonly operated at the temperature of 200–320 °C under an inert environment [3,19,26,27], and it is considered one of the most promising pretreatment methods prior to biomass gasification [28,29,30,31]. Typically, 70–80% of the initial mass is retained as a solid product (torrefied biomass), and 85–90% of the initial energy is maintained by means of biomass torrefaction [16,32]. Torrefaction has been widely applied as a pretreatment for biomass gasification, and the characteristics, such as chemical compositions [19], porosity structure [33], and ash content [34], are remarkably changed, which can strongly affect the gasification performance [28,35]. For example, Couhert et al. [36] found that torrefied wood could produce 7% of more H_2_ and 20% of more CO than untreated wood during biomass gasification, and the biochar achieved from torrefied wood was observed to be less reactive than that gained from untreated wood. Dudyński et al. [23] reported that biomass torrefaction could enhance syngas quality and cold gas efficiency during biomass gasification. With regard to tar formation behaviors, the torrefaction introduced before biomass gasification reduced the total tar content by over 50% [37]. Marcello et al. [38] also noted that the carbon conversion efficiency (CCE) and cold gas efficiency (CGE) were both enhanced because of the promotion of the grindability of the torrefied wood, and the decreased volatile matter content could reduce the tar content in the syngas. 

To date, steam gasification emerging as a more advanced gasification technology, is usually applied to produce a combustible gas containing H_2_-rich syngas and reduce tar formation, and a variety of studies have conducted the steam gasification of diverse biomass for syngas production [39,40,41], but few studies have recently determined the effect of torrefaction pretreatment on steam gasification of biomass [28,42,43,44]. Sarker et al. [43] reported that the highest cumulative gas yield (24.4 mol/kg), syngas yield (22.1 mol/kg), carbon conversion efficiency (85.1%), and lower heating volume of the gas (2.40 kJ/Nm^3^) were achieved from steam gasification of torrefied canola residue pellets at 800 °C with the equivalence ratio of 0.4. Moreover, Singh and Yadav [42] observed that the H_2_ yield and syngas yield ranged from 0.60 to 2.15 m^3^/kg and from 0.95 to 3.49 m^3^/kg, respectively, from steam gasification of mixed food waste torrefied at varying temperatures. 

The limited above-mentioned studies have integrated torrefaction pre-treatment with steam gasification for gas production and tar reduction. Furthermore, there is no research focused on torrefaction pretreatment followed by steam gasification to convert agricultural biomass into gas. It should be noted that carbonization, as more rigorous than torrefaction, has seldom been combined with steam gasification for H_2_-enriched syngas production and tar reduction. To the best of our knowledge, this is the first study emphasizing steam gasification of torrefied/carbonized wheat straw. The objective of this study was to investigate the impact of torrefaction/carbonization pretreatment on steam gasification performance in terms of gas formation behaviors and tar reduction performance. 

Herein, the torrefaction/carbonization pretreatment of wheat straw combined with steam gasification was performed to achieve high-yield and clean H_2_-enriched syngas with low tar content.

## 2. Materials and Methods

### 2.1. Materials

Wheat straw (WS) was chosen as presentative agricultural biomass supplied by the Shangzhuang experimental station of the China Agricultural University. Wheat straw (WS) was oven-dried at 105 °C to remove the physically bound moisture, and then the dried WS was pulverized with a particle size of approximately 20–40 mesh. The proximate analysis, ultimate analysis, and higher heating value (HHV) of WS are listed in Table 1.

### 2.2. Torrefaction/Carbonization of Wheat Straw

As displayed in Figure 1, torrefaction/carbonization of WS was operated in a horizontal tubular furnace. The reactor was initially purged with Ar at a flow rate of 100 mL/min for 30 min. WS was then torrefied in the horizontal tubular reactor at varying temperatures (220, 240, 260, 280, and 300 °C) with a consistent residence time of 1 h. Additionally, carbonization of WS was also conducted in the horizontal tubular reactor at 400 and 500 °C by using the same experimental procedures. The torrefied/carbonized WS feedstocks were labeled as WS-*X*, and *X* represents the treated temperature. 

### 2.3. Steam Gasification of Torrefied/Carbonized Wheat Straw

As shown in Figure 2, the experimental apparatus was made of a gas/steam supplying system, a vertical tubular furnace with a temperature controller, a condensation system, gas collection system, and so forth. The quartz reactor was assembled in the vertical furnace. In all tests, raw or torrefied/carbonized WS with a consistent weight (1.5 g) was preloaded in a quartz sampling basket, and the bottom of the basket was porous for the emission of volatiles. Prior to the test, the quartz basket was hung outside of the furnace. The gasification system was first purged with Ar at a flow rate of 100 mL and then preheated to 250 °C. When reaching 250 °C, the Ar flow rate was stopped, and the sampling basket was rapidly pushed into the gasification system to initiate the experiment. At the same time, water was pumped into the reactor using a syringe pump at a rate of 0.2 mL/min, and steam was generated once the water flowed into the reactor. Before terminating each experiment, the system was continuously elevated to the desired temperature (700–900 °C) from 250 °C at 10 °C/min and maintained for a certain time to ensure the total residence time equaled 75 min. Finally, the gas products from the gasification system passing through the condensation system were collected for further GC analysis. The condensed tar was trapped in a condensing system, and we used the same ethyl acetate (30 mL) to wash the tar stuck to the condensation system, and the ethyl acetate-soluble tar phase was collected for further GC/MS. For each test, several repeated runs were conducted to assure the repeatability of the results. 

### 2.4. Analytical Techniques

The elemental compositions (C, H, N, and S) of raw and torrefied/carbonized feedstocks were measured by an elemental analyzer (Elementar Vario ELIII, Hanau, Germany). Approximately 20 mg of the sample was first prepared as a pellet with a diameter of 0.5 mm using tin foil. Then, the samples were tested using CHNS mode. The amounts of moisture, volatile matter, and ash were determined by utilizing an automatic industrial analyzer (YX-GYFX 7705B, U-Therm, Changsha, China). The fixed carbon was measured by the subtraction method. The functional groups on terrified/carbonized WS feedstocks were characterized by Fourier transform-infrared (FTIR) spectroscopy (Spectrum 400, PerkinElmer, Waltham, MA, USA). The spectra were recorded at a range of 400–4000 cm^−1^.

The chemical components in tar were identified by a GC/MS(QP2010 SE, Shimadzu, Kyoto, Japan) with an RTX-5MS (30 m × 0.25 mm × 0.25 µm) capillary column. The GC was first heated to 60 °C for 5 min and then further heated to 270 °C at a rate of 5 °C/min and kept at 270 °C for 5 min. The injection volume was 1 µL, and the injection temperature was set at 300 °C. The interface temperature was changed to 280 °C, and the ion source temperature was maintained at 200 °C for the mass selective detector. High-purity helium (99.999%) was utilized as the carrier gas at a stable flow rate of 3 mL/min, and the split radio was 1:10. The identification of chemical compounds in tar was determined based on the NIST database of MS spectra library. All the GC/MS measurements were operated in triplicate to assure reproducibility. 

All gas products were collected in a 1 L gas bag and then offline analyzed by a Shimadzu GC-2014C gas chromatography (GC, Shimadzu Corp., Kyoto, Japan) with a thermal conductivity detector (TCD). The GC was maintained at 150 °C for 5 min and then further heated to 200 °C at a rate of 50 °C/min and kept at 200 °C for 10 min. A standard gas mixture (i.e., H_2_, CO, CH_4_, CO_2_, C_2_H_4_, C_2_H_6_, C_3_H_6_ and C_3_H_8_) was used to calibrate the proportion (vol%) of gaseous fractions. The gas fractions (≥C_3_) were not identified or negligible in this study. All GC measurements were also conducted in triplicate to ensure reproducibility.

### 2.5. Data Analysis

The higher heating value (*HHV*) of raw and torrefied/carbonized WS feedstocks was determined by the modified Dulong formula as follows [45]:(1)HHV(MJ⁄kg)=0.3578×C+1.1357×H−0.0845×O+0.0594×N+0.119×S

The energy yield (*EY*) was defined as follows [28]:(2)$$EY=\frac{HHV_{Treated}\times m_{Treated}}{HHV_{WS}\times m_{WS}}\times 100\%$$
where $HHV_{Treated}$ is the higher heating value of torrefied/carbonized WS, $HHV_{WS}$ is the higher heating value of raw WS, $m_{Treated}$ (g) is the mass of torrefied/carbonized WS, and $m_{WS}$ is the mass of raw WS. 

The gas yield ($Y_{i}$, mmol/g) is defined as the mole of gas *i* produced by 1 g of raw or torrefied/carbonized WS feedstock [28]:(3)$$Y_{i}=\frac{V}{22.4\times m}\times x_{i}$$
where *x_i_* (vol%) is the concentration of gas component *i* (H_2_, CO, CH_4_, CO_2_, C_2_H_4_ and C_2_H_6_) measured by GC; *V* (mL) is the collected cumulative gas volume; *m* (g) represents the mass of raw or torrefied/carbonized WS feedstock supplied for the gasification process.

The syngas yield (*Y_syngas_*, mmol/g) is calculated as follows [28]:(4)$$Y_{syngas}=\frac{V}{22.4\times m}\times \left(x_{H_{2}}+x_{CO}\right)\;$$

Syngas purity (*Ps*, %) is denoted as the total mole fraction of H_2_ and CO in the gaseous product [28]:(5)$$P_{s}=x_{H_{2}}+x_{CO}$$

The lower heating value (LHV) of the gaseous products was evaluated using the following equation [46].
(6)LHV(MJNm3)=0.126×CO+0.108×H2+0.358×CH4+0.665×CnHm
where CO, H_2_, CH_4_ and C_n_H_m_ (≥C_2_) are the volumetric fractions of the gas compositions. 

To quantify the conversion properties of raw WS or torrefied/carbonized WS during the steam gasification process, the carbon conversion efficiency (CCE, C%) [28] and cold gas efficiency (CGE, %) [43] were adopted to determine the thermal conversion degree for gas production.
(7)$$CCE=\frac{12}{22.4}\times \frac{\sum V_{i}}{m\times f_{C}}\times 100\%$$
where *V_i_* is the volume (L) of each compound *i* (*i*: CO, CH_4_, CO_2_, C_2_H_4_, and C_2_H_6_); *f_c_* (C%) is the weight fraction of carbon in the raw or torrefied/carbonized WS feedstock. 

The cold gas efficiency (CGE, %) of the steam gasification is expressed as the ratio of the calorific value of the gas products to the calorific value of feedstock gasified in the system:(8)$$CGE=\frac{LHV_{gas}\times V}{LHV_{feedstock}\times m}\times 100\%\;$$
where *LHV_feedstock_* is the lower heating value of raw or torrefied/carbonized WS feedstock; *LHV_gas_* is the lower heating value of generated gas (MJ/m^3^), which is calculated by Equation (6).

## 3. Results and Discussion

### 3.1. Properties of Torrefied/Carbonized Wheat Straw

Note that the color of torrefied/carbonized WS became darker with the increase in the treated temperature from 220 to 500 °C (see Appendix A), implying that the carbonization severity of WS was associated with the treated temperature. These outcomes were further verified by the FTIR analysis, as shown in Appendix A. The peak intensities were obviously reduced when torrefaction was conducted for raw WS, and the peak intensities gradually declined as the increase in pretreatment temperature. These findings suggested that the cellulose and hemicellulose fractions in WS were significantly decomposed by torrefaction/carbonization [16,30,31].

The mass yield, energy yield, and HHV of WS torrefied/carbonized at various temperatures are demonstrated in Figure 3. Obviously, the treated temperature strongly affected the mass and energy yield of torrefied/carbonized WS. The mass and energy yields of torrefied/carbonized WS gradually declined with the increasing treated temperature. The mass yield was high (>84%) when the torrefaction temperature was kept at 220 °C, while it dramatically dropped with further increasing the treated temperature. For instance, the solid yield was reduced to only 44.58% at 300 °C, which was comparable with the result reported elsewhere [28]. When the treated temperature was elevated to the carbonization temperature range, i.e., 400 °C and 500 °C, the mass yield declined to ~35%, suggesting the relatively entire carbonization of WS.

The HHV of the torrefied/carbonized WS feedstocks is also displayed in Figure 3. The HHV of the torrefied/carbonized WS was gradually augmented with the increase in treated temperature, reaching a plateau at 280–300 °C. Some relevant studies also reported that the constant maximums were achieved at 280–330 °C [28,47], and a further incline in the treated temperature resulted in a slight drop in HHV, which was ascribed to the release of hydrocarbon gases [28]. In addition, the energy yield of the torrefied/carbonized WS is demonstrated in Figure 3. The energy yield continuously declined as the treated temperature was elevated, presenting a remarkable decreasing tendency when the treated temperature was ≥260 °C. When the treated temperature was equal to or larger than 400 °C, the energy yield of carbonized WS started to decline slightly. Similarly, the energy loss was extremely high at the high treated temperatures, e.g., 56.51% at 500 °C. Notably, the energy yield of all torrefied/carbonized WS feedstocks was evidently higher than the mass yield under the same torrefaction/carbonization condition, suggesting that torrefaction/carbonization is a viable way for energy densification of biomass. Even though the fuel properties of the torrefied/carbonized WS were enhanced with the increase in the treated temperature, an optimal treated temperature should be selected because a higher treated temperature not only causes the increase in energy input but also results in more energy loss. Based on the above outcomes, a treated temperature that remained at 280 °C was appropriate considering both the fuel properties and energy input. 

The proximate and ultimate analysis of the torrefied/carbonized WS at various treated temperatures were conducted, and the results are demonstrated in Table 1. The increase in torrefaction/carbonization severity strongly affected the content of volatile matter and fixed carbon content. The release of volatile matter modestly took place when the treated temperature was ≤240 °C, gradually declining from 72.34% for raw WS to 61.48 for WS-240, yet further increasing the treated temperature to 500 °C resulted in the considerable decrease in volatile matter to 11.56%. Conversely, the fixed carbon content presented an increasing tendency from 14.28% for raw WS to 61.26% for WS-500, suggesting that the devolatilization almost entirely happened at the carbonized temperature of 500 °C. Owing to the release of the volatile matter during the torrefaction/carbonization pretreatment, the ash content was gradually aggregated in the treated WS with the increase in treated temperature. As for the ultimate analysis, it was clearly found that the torrefaction/carbonization pretreatment significantly increased C content from 43.09 to 61.36% but reduced H content from 5.31 to 2.02%. More importantly, the O content was reduced from 42.33% for raw WS to 11.10% for WS-500. These observations evidenced that the higher pretreatment severity led to more promotions of dehydration and deoxygenation reactions, contributing to the increase in carbonization degree. 

The H/C and O/C atomic ratios of the torrefied/carbonized WS feedstocks were further compared with other solid fuels in the Van Krevelen diagram in Figure 4. With the increase in torrefaction/carbonization severity, the atomic ratios of H/C and O/C were graduated reduced, ca. from 1.27 and 0.63 for WS-220 to 0.39 and 0.14 at for WS-500. Notably, the atomic ratios of H/C and O/C of WS-220 and WS-240 almost showed a linear correlation with a scope of 2, indicating that dehydration was the dominant reaction taking place during the torrefaction at the low temperatures [28]. With the increase in the pretreated temperature, the decarboxylation and decarbonylation reactions were intensified, thereby making the corrections of H/C and O/C atomic rations nonlinear. These findings suggested that WS torrefaction was a mild carbonization process accompanied by dehydration and deoxygenation reactions. The main devolatilization reactions during torrefaction usually contained dehydration, decarboxylation, and decarbonylaiton reactions [3]. The H/C and O/C atomic ratios of the pretreated WS at the high treated temperatures were even close to coal and anthracite, implying the improvement of the fuel properties of these torrefied/carbonized WS. Accordingly, it was inferred that the variations in the atomic H/C and O/C ratios were attributed to a sequence of reactions taking place during the torrefaction/carbonization of WS. 

### 3.2. Influence of Operational Temperature on Steam Gasification

#### 3.2.1. Gas Formation Behaviors 

The gasification temperature is a very important factor that affects gasification performance and reaction rates. The gas formation behaviors from steam gasification of WS as a function of gasification temperature, varying from 700 to 900 °C, are demonstrated in Figure 5. Briefly, the gas products obtained at the gasification temperature were mainly composed of H_2_, CO, CH_4_, and CO_2_. As shown in Figure 5a, the volumetric concentration of H_2_ clearly went up from 34.55 to 47.97 vol% as the gasification temperature was increased from 700 to 900 °C, suggesting that a higher gasification temperature was in favor of H_2_ formation reactions such as water gas reactions and Water-gas shift (WGS) reaction. Additionally, H_2_ gradually became the most predominant fraction when the gasification was ≥750 °C. The CO concentration first declined to 29.72 vol% when the gasification temperature was elevated to 800 °C, while the CO concentration became stable with further increasing the gasification temperature to 900 °C. It was noted that the CH_4_ concentration exhibited a descending trend from 11.14 to 6.08 vol%, that was because the endothermic steam methane reforming (SMR) reaction was accelerated as the gasification temperature was increased, contributing to the more formation of syngas. Importantly, the increase in gasification temperature was good for the augmenting of syngas purity, indicating that the decrease in CO concentration was offset by the improvement of H_2_ content. As for the syngas quality, the ratio of H_2_/CO molar ratio is an essential index for further use [28]. Owing to the high content of H_2_, the syngas purity peaked at 77.70%, and the H_2_/CO ratio of 1.61 was gained at 900 °C, which was suitable as the syngas precursor for the downstream synthesis of chemicals or fuels. 

Upon the increase in gasification temperature, a considerable increase in gas yields was achieved, as demonstrated in Figure 5b. It was noteworthy that the elevation of gasification temperature in the range of 700–900 °C was beneficial for the increase in the cumulative gas yield. As the gasification temperature was promoted, the endothermic reactions, including water gas reactions [48] and tar reforming reactions [20], could be improved, contributing to the H_2_ formation. Correspondingly, both the H_2_ and syngas yield was dramatically enhanced with the increase in gasification temperature. As such, the CO yield gradually increased with the increase in gasification temperature, suggesting that the water gas reactions were always more dominant than the WGS reaction [28], giving rise to both the increases in H_2_ and CO yields. The maximum cumulative gas yield, H_2_ yield, and syngas yield were all achieved when the gasification temperature was kept at 900 °C, reaching 75.89 mmol/g, 36.41 mmol/g, and 58.97 mmol/g, respectively. It was proven that the steam reforming reactions, including the WGS reaction and steam–methane-reforming (SMR) reaction, were responsible for the increase in gas yields [39,49].

#### 3.2.2. Carbon Conversion Efficiency and Cold Gas Efficiency

The carbon conversion efficiency and cold gas efficiency were also used to demonstrate the thermal conversion degree of WS for gas production. As expected, the two indicators revealed ascending tendencies in the range of 700–900 °C, as shown in Figure 6. That was because the higher gasification temperature facilitated the gasification reaction rates, resulting in more generation of gas productions. More specifically, the CCE and CGE results obtained at 700 °C were only 55.92 C% and 65.71%, respectively, while the effluent gas product obtained at 900 °C showed the highest CCE (nearly 100 C%) and CGE (120.41%). It was inferred that the gasification temperature of 900 °C was preferred because of the complete carbon conversion into gas fractions. Similarly, other studies also found that the steam gasification of biomass at 900 °C is optimal for syngas production [50,51]. Thus, the steam gasification at 900 °C was used as the preferred condition for the following steam gasification operations. 

### 3.3. Coupling of Torrefaction/Carbonization and Steam Gasification

#### 3.3.1. Gas Formation Behaviors

To evaluate the effect of torrefaction/carbonization severity on gas and tar formation behaviors from the steam gasification process. The torrefied WS at 220–300 °C and carbonized WS at the pyrolysis temperatures (i.e., 400 and 500 °C) were subjected to the steam gasification process for syngas production. The evolved gas compositions, gas yields, CCE, and CGE from steam gasification of raw WS and WS torrefied/carbonized at varying temperatures are tabulated in Table 2. As WS was pretreated from 220 to 500 °C, the H_2_ concentration steadily went up from 47.97 for raw WS to 54.21 vol% for WS-500, whilst the volumetric concentrations of CH_4_ and C_n_H_m_ gradually declined to very low levels for WS-500. The reduction in the concentrations of CH_4_ and C_n_H_m_ was due to the pre-decarbonylation, pre-demethylation, and pre-removal of light oxygenates during the torrefaction/carbonization process [16]. The dramatic decrease in CH_4_ and C_n_H_m_ would, in turn, result in the increasing concentrations of other gaseous fractions. However, torrefaction/carbonization severity did not present a strong effect on the CO concentration, slightly fluctuating at 30 vol%.

Generally, the enhancement of the pretreatment severity gave rise to the continuous increase in syngas purity from 77.70% for raw WS to 85.59 vol% for WS-500. The H_2_/CO molar ratio of torrefied/carbonized WS feedstocks was enhanced as compared to that of raw WS. The highest H_2_/CO molar ratio (1.83) was observed from steam gasification of WS torrefied at 280 °C. Further, significantly increasing the pretreated temperature to 400 and 500 °C would not contribute to the increase in the H_2_/CO molar ratio.

The cumulative gas yield, H_2_ yield, and syngas yield are also illustrated in Table 2. The cumulative gas yield, H_2_ yield, and syngas yield from steam gasification of raw WS were 75.89 mmol/g, 36.41 mmol/g, and 58.97 mmol/g, respectively; while they were gradually lifted as the pretreated temperature was elevated from 220–280 °C, maximizing at 97.47 mmol/g, 50.92 mmol/g, and 78.77 mmol/g respectively. Compared with the raw WS, WS-280 enhanced the gas yields by 28.43%, 39.85%, and 33.58%, respectively. As the pretreated temperature increased from 280 to 400 °C, the cumulative gas yield apparently dropped. This outcome was ascribed to the devolatilization, polycondensation, and carbonization of WS during the torrefaction/carbonization process, giving rise to the generation of more biochar with lower reactivity in the downstream gasification [16,29]. Noteworthily, the cumulative gas yield, H_2_ yield, and syngas yield peaked at 102.68 mmol/g, 55.66 mmol/g, and 87.89 mmol/g, respectively, from steam gasification of WS-500, which was possibly due to more water gas reactions occurring during the steam gasification process, as indicated by the higher C and fixed carbon content in WS-500 (see Table 1). These findings affirmed that torrefaction/carbonization is an effective pretreatment approach that can be integrated with steam gasification for boosting H_2_-rich syngas production. 

#### 3.3.2. Carbon Conversion Efficiency and Cold Gas Efficiency

The CCE and CGE values are also the dominant parameters for assessing the gasification performance of solid fuels [13]. The CCE and CGE values from steam gasification of raw WS and WS treated at various temperatures are also demonstrated in Table 2. It was found that WS-200 showed comparable CCE and CGE results with raw WS, indicating that the carbon content in WS-220 was completely transformed and reformed into gas product. However, the torrefied WS feedstocks except WS-280 experienced decreasing tendencies for the CCE and CGE results, as the torrefied temperature was increased from 220–300 °C. It was worth noting that WS-280 presented decent CCE (90.33 C%) and CGE (109.24%) among these torrefied WS feedstocks. In comparison, the CCE and CGE results of WS-280 were much higher than that of WS-260 and WS-300. These findings suggested that torrefaction at 280 °C could enhance the CCE and CGE, but excessive torrefaction severity resulted in a decrease in the CCE and CGE. These trends were in agreement with the study reported elsewhere [13]. On the other hand, the carbonized WS feedstocks (i.e., WS-400 and WS-500) showed the lowest CCE and CGE results among these pretreated WS feedstocks. That was because the higher pretreated severity caused the more polymerized and stable biochar feedstocks, which possessed the lower reactivity during the steam gasification process. Given the evolved gas compositions, gas yields, CCE, and CGE result from the steam gasification of raw WS and torrefied/carbonized feedstocks, as well as energy input, WS-280 was the ideal and promising feedstock for steam gasification.

To further evaluate the gasification performance from the integrated process, the gas formation behaviors achieved in this study should be compared with the results obtained from the representative studies. Marcello et al. [38] found that steam-oxygen circulating fluidized bed gasification of mixed wood torrefied at 300 °C contributed to an increased gas yield and quality, with a gas yield of 75.89 mmol/g, H_2/_CO molar ratio of 2.1, CCE of 92.5%, and CGE of 56.0%. Similar to our work, Sarker et al. [43] recently investigated the effect of torrefaction by microwave treatment on steam gasification of canola residue at 800 °C, and the total gas yield, syngas yield, CCE were 18.4 mmol/g, 16.5 mmol/g, and 61.4%, respectively. Moreover, Chew’s group [52] introduced torrefaction pretreatment to promote the oil palm biomass properties prior to gasification, and it was observed that mesocarp fibers torrefied at 280 °C as the feedstock for steam gasification gained a higher CCE (55.74%) and CGE (61.24%) than palm kernel shells pretreated under the same condition. Therefore, it is affirmed that torrefaction/carbonization integrated with steam gasification to valorize wheat straw in this study was superior to these representative studies in terms of gas yields, CCE, CGE, etc. In a word, integrating torrefaction/carbonization with steam gasification is promising to valorize wheat straw into high-yield H_2_-enriched syngas.

#### 3.3.3. Tar Elimination Performance

To reveal the effect of torrefaction/carbonization on tar formation from steam gasification of torrefied/carbonized WS, the total ion chromatograms originating from steam gasification of raw and torrefied/carbonized WS feedstocks are illustrated in Figure 7. Even though the relative peak area of each compound cannot represent the actual tar yield, it can provide insight into the effect of torrefaction/carbonization severity on the variation tendency of tar yield from the steam gasification process. As displayed in Figure 7a, the constituents of tar were classified into monocyclic aromatic hydrocarbons (MAHs), polycyclic aromatic hydrocarbons (PAHs), phenolics, and other aromatics, and the most dominant components in tar were PAHs and MAHs. As demonstrated in Figure 7b, WS-220, WS-240, and WS-260 did not present the contribution to tar elimination; that was because WS torrefied at the low temperature range (220–260 °C) was more difficult to be reformed, resulting in more tar formation. This outcome was in agreement with the study reported elsewhere [28]. However, the tar intensity gradually declined with the promotion of treated temperature from 260–500 °C, which was possibly due to the fact the higher carbonization degree caused by the higher torrefied/carbonized severity made the torrefied/carbonized WS less reactive and generate less volatiles to form tar. Apparently, the carbonized WS feedstocks (WS-400 and WS-500) exhibited very low tar intensities in comparison with both raw WS and torrefied WS feedstocks, and the lowest tar content was achieved from steam gasification of WS-500. It was assumed that the carbonization of hemicellulose and cellulose at 400 and 500 °C proceeded with severe polycondensation and carbonization reactions to generate carbonized intermediates with aromatic structures [53]. Furthermore, the pre-removal of tar precursors (such as holocellulose-derived oxygenates and phenyl units) was successfully fulfilled during the high-severity torrefaction and carbonization of WS [16]. It was concluded that the pre-removal of holocellulose-derived oxygenates and lignin-derived phenols during biomass torrefaction/carbonization could lower the generation of tar precursors during subsequent steam gasification, contributing to the reduction in tar content. 

## 4. Conclusions

In summary, the impact of biomass torrefaction/carbonization on the H_2_-enriched syngas production and tar formation from the steam gasification process was investigated in detail. First, torrefaction/carbonization of WS was conducted at varying temperatures of 220–500 °C. It was found that the treated temperature strongly affected the mass and energy yield of torrefied/carbonized WS. The HHV of the torrefied/carbonized WS reached a plateau at 280–300 °C. It was inferred that a treated temperature of 280 °C was appropriate considering both the fuel properties and energy input. Thereafter, the steam gasification process as a function of gasification temperature varying from 700–900 °C was performed. It was observed that the maximum cumulative gas yield, H_2_ yield, and syngas yield were all achieved when the gasification temperature was kept at 900 °C. Afterward, the torrefied/carbonized WS feedstocks were subjected to the optimized steam gasification process. It was found that the H_2_ concentration steadily went up to 54.21 vol% for WS-500, and the enhancement of the pretreatment severity resulted in the increase in syngas to 85.59 vol% for WS-500. Yet, the highest H_2_/CO molar ratio (1.83) was observed for WS-280. Moreover, WS-280 presented decent CCE (90.33 C%) and CGE (109.24%) among these torrefied WS feedstocks. Notably, the cumulative gas yield, H_2_ yield, and syngas yield peaked at 102.68 mmol/g, 55.66 mmol/g, and 87.89 mmol/g, respectively, from steam gasification of WS-500. Compared with raw and torrefied WS feedstocks, the carbonized WS feedstocks, especially WS-500, revealed a lower content. As a result, integrating biomass torrefaction/carbonization with steam gasification would provide a promising strategy for enhancing H_2_-enriched syngas production whilst minimizing tar formation.

## Figures and Tables

**Figure 1 ijerph-19-10475-f001:**
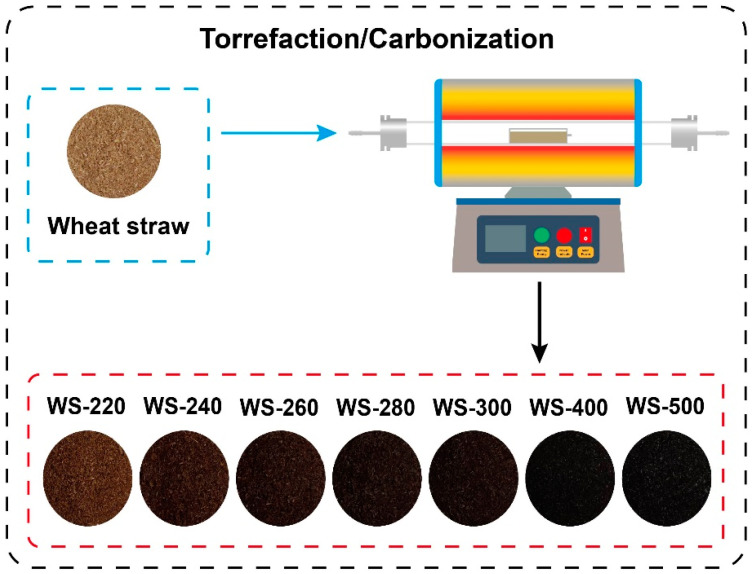
Schematic diagram of torrefaction/carbonization of raw WS.

**Figure 2 ijerph-19-10475-f002:**
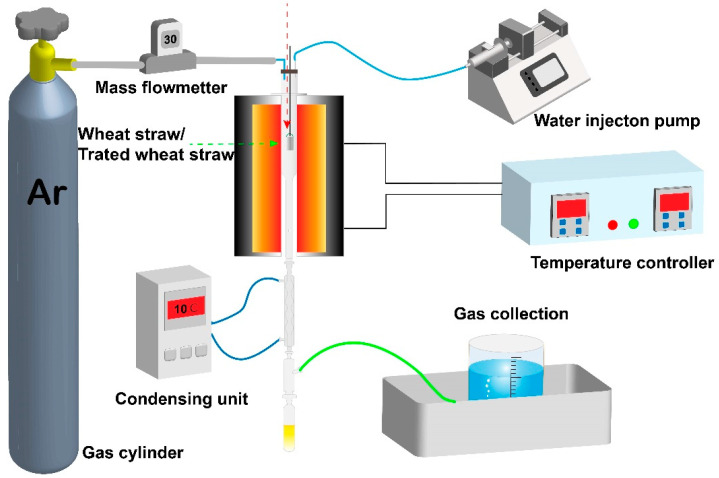
Schematic diagram of steam gasification of raw WS and torrefied/carbonized WS.

**Figure 3 ijerph-19-10475-f003:**
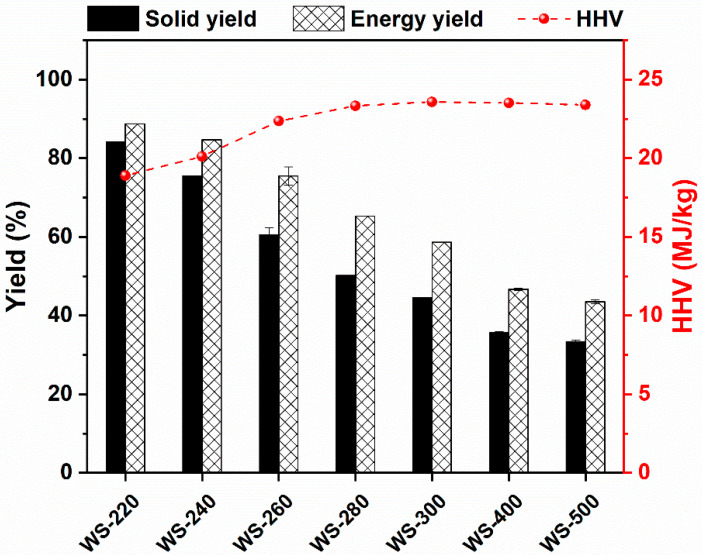
Mass yield, energy yield, and energy density of WS treated at various temperatures.

**Figure 4 ijerph-19-10475-f004:**
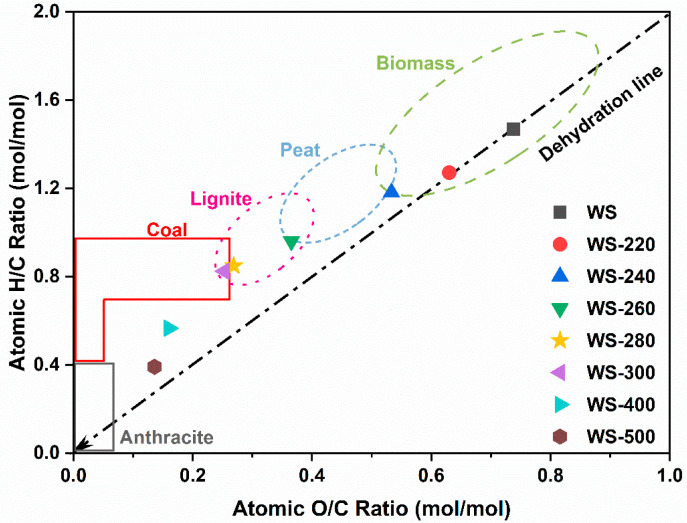
Van Krevelen diagram of raw WS and WS treated at various temperatures.

**Figure 5 ijerph-19-10475-f005:**
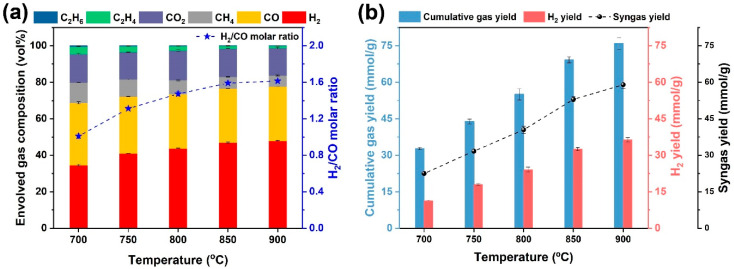
Gas formation behaviors from steam gasification of raw WS with regard to gasification temperature: (**a**) evolved gas composition and H_2_/CO molar ratio and (**b**) cumulative gas yield, H_2_ yield, and syngas yield.

**Figure 6 ijerph-19-10475-f006:**
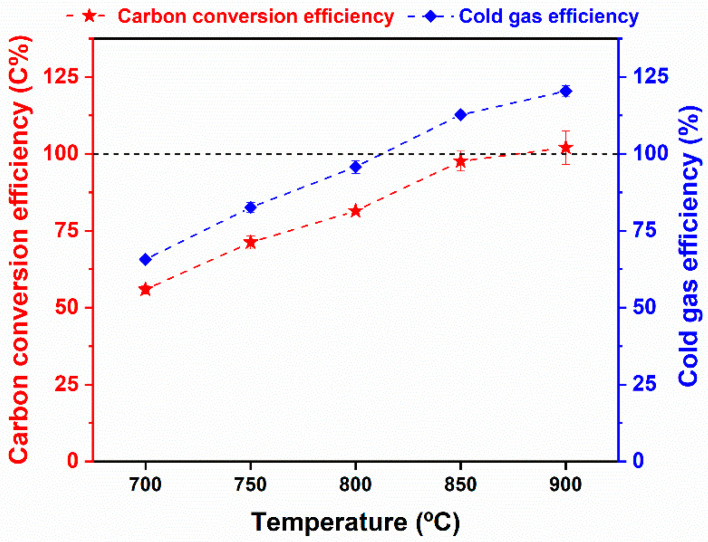
The carbon conversion efficiency and cold gas efficiency from steam gasification of raw WS as a function of gasification temperature.

**Figure 7 ijerph-19-10475-f007:**
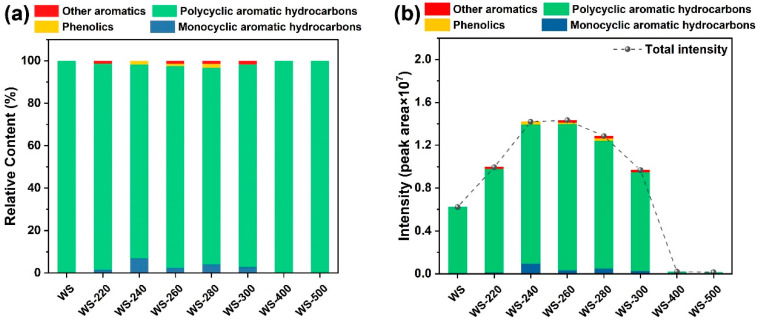
Tar elimination behaviors from steam gasification of raw WS and WS treated at varying temperatures: (**a**) relative contents of aromatics in tar and (**b**) intensities of aromatics in tar.

**Table 1 ijerph-19-10475-t001:** Proximate and ultimate analysis of raw and torrefied/carbonized wheat straw.

Characteristics	WS	WS-220	WS-240	WS-260	WS-280	WS-300	WS-400	WS-500
*Proximate analysis (wt%)*								
Moisture	4.97	2.36	3.64	2.50	1.17	2.83	3.33	2.79
Volatile	72.34	66.73	61.48	50.09	36.13	33.67	19.56	11.56
Fixed carbon ^a^	14.28	21.58	24.62	33.97	45.55	45.98	53.27	61.26
Ash	8.41	9.33	10.26	13.44	17.15	17.52	23.84	24.39
*Ultimate analysis (wt%)*								
C	43.09	46.17	48.62	54.61	56.78	57.51	59.50	61.36
H	5.31	4.93	4.82	4.40	4.05	3.98	2.83	2.02
O ^b^	42.33	38.76	34.53	26.58	20.33	19.38	12.70	11.10
N	0.75	0.64	0.90	0.80	1.09	1.14	0.91	0.88
S	0.11	0.17	0.87	0.17	0.60	0.48	0.22	0.25
HHV (MJ/kg)	17.93	18.90	20.11	22.36	23.33	23.58	23.51	23.39

^a^ Determined by difference. ^b^ O (%) = 100% − (C (%) + H (%) + N (%) + S (%) + Ash (%)).

**Table 2 ijerph-19-10475-t002:** Gas formation behaviors from steam gasification of torrefied/carbonization wheat straw at 900 °C.

Feedstocks	Gas Yields (mmol/g_feedstock_)			Gas Concentration (vol%)						H_2_/CO	Syngas Purity (%)	CCE (C%)	CGE (%)
	Gas	H_2_	Syngas	H_2_	CO	CH_4_	CO_2_	C_2_H_4_	C_2_H_6_				
WS	75.89	36.41	58.97	47.97	29.73	6.08	15.07	1.15	0.01	1.61	77.70	101.99	120.41
WS-220	79.61	38.19	61.91	47.97	29.79	5.78	15.09	1.34	0.02	1.61	77.76	101.76	119.25
WS-240	79.61	39.32	63.22	49.39	30.02	5.33	14.07	1.17	0.01	1.64	78.20	96.63	110.58
WS-260	87.80	44.66	70.53	50.86	29.47	4.98	13.88	0.80	0.01	1.73	80.23	88.66	105.97
WS-280	97.47	50.92	78.77	52.24	28.57	4.58	14.09	0.52	0.00	1.83	80.81	90.33	109.24
WS-300	91.52	47.89	74.91	52.33	29.53	4.65	12.98	0.51	0.01	1.77	81.86	84.04	102.73
WS-400	95.24	51.33	79.95	53.89	30.06	3.18	12.78	0.09	0.00	1.79	83.95	81.41	100.75
WS-500	102.68	55.66	87.89	54.21	31.38	1.93	12.47	0.01	0.00	1.73	85.59	81.99	105.33

## Data Availability

Not applicable.

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
