# Peer review of "Steam Gasification of Torrefied/Carbonized Wheat Straw for H2-Enriched Syngas Production and Tar Reduction"

_ijerph, 2022, doi:10.3390/ijerph191710475_

Round 1

Reviewer 1 Report

This scientific investigation is very relevant since pre-treatment methods for improving biomass gasification are required. The methodology is well explained and the results well presented. Conclusions are supported by the results. The english should be checked an corrected to eliminate the last mistakes.

Reviewer 2 Report

General Comments

The study investigates the effect of torrefaction and carbonization (being biomass heat treatment methods) on the performance of wheat straw during gasification. The study is worth reading and contributes to knowledge. However some comments need to be taken to improve the look of the manuscript:

1. There are too many grammatical errors. The entire manuscript should be read and corrected for errors.

2. Improve citations throughout; e.g. L.64 should be corrected to “Couhert et al. [36] found…” Correct references in L.67,82,86,etc. L.80 refers to Huang’s group; this should be properly cited.

3. Use appropriate degree symbol.

4. Define abbreviations, acronyms and symbols at the first use.

5. Be consistent with acronyms (e.g. GC/MS or GC-MS)

Title

6. The title needs to be rephrased to include “Wheat Straw” which is considered as the object of study and to which the authors ascribe the novelty of the study.

Abstract

7. “Unraveled” should be changed to “investigated” or a more appropriate word.

8. L.14-16: The sentences herein are not clear. Gasification was carried out between 700 and 900°C. This is what ought to be stated (unless I missed the point).

Introduction

9. L.55: As far as I know, fast pyrolysis optimizes liquid (bio-oil/pyro-oil) production. Could this be listed as a pretreatment approach since gasification converts solids directly into gas?

10. L.56-57: Torrefaction is described as low-temperature carbonization. Since carbonization is a unique process having its parameters, I think the statement ought to be rephrased by replacing “carbonization” with “heat treatment” or a suitable description.

11. L.89-92: The sentence is confusing. Recheck and probably use shorter sentences.

12. L.100-108: Methodology need not be described in the introduction.

Experimental section

13. The title of the section should be changed. It should read “experimental methods” or more appropriately – “materials and methods”

14. L113: How is it possible to air-dry at 105°C? I suppose you mean oven-dried.

15. L.114-117: Table 1 presents results. This should be moved to appropriate section in results and discussion.

16. L.122-124: You should rather state that carbonization was carried out at the specified temperatures. The equipment used for carbonization was not described and is not easily deduced.

17. As with Figure 1 (for gasification), could a schematic diagram be provided for torrefaction and carbonization? Authors may decide the necessity of this.

18. L.138: State the desired temperature.

19. Figure 1 is not fully labelled.

20. Cite the references from where the equations stated for data analysis were obtained.

Results and discussion

21. Could Figures S1 and S2 be included in the main manuscript? Authors may want to consider this.

22. L.223: What is meant by “very stable level”?

23. L.260,271: I believe you mean dehydrogenation

24. Section 32 should include “…of raw WS” because that is what is being considered

25. Figure 5 shows CCE and CGE are >1100% at some points. Could an explanation be given? These results have not been properly discussed by comparison with other related studies.

26. L.364-365: You should state the exact gas yields which each value represents (due to the construct of the sentence).

27. Figure 6: No explanation was given for why the tar intensity increased between 220 and 260 °C.

Reviewer 3 Report

My commnets are attached as a PDF document

Round 2

Reviewer 3 Report

Authors have developed a massive effort to review their original work. I think they have considered most of my suggestions and I have no additional comments. I think this work now reaches quality standards to be published in this journal. My suggestion is to accept it in its current form.